# Deep Transcranial Magnetic Stimulation Affects Gut Microbiota Composition in Obesity: Results of Randomized Clinical Trial

**DOI:** 10.3390/ijms22094692

**Published:** 2021-04-29

**Authors:** Anna Ferrulli, Lorenzo Drago, Sara Gandini, Stefano Massarini, Federica Bellerba, Pamela Senesi, Ileana Terruzzi, Livio Luzi

**Affiliations:** 1Department of Endocrinology, Nutrition and Metabolic Diseases, IRCCS MultiMedica, 20099 Milan, Italy; anna.ferrulli@multimedica.it (A.F.); stefano.massarini@multimedica.it (S.M.); pamela.senesi@unimi.it (P.S.); ileana.terruzzi@unimi.it (I.T.); 2Department of Biomedical Sciences for Health, University of Milan, 20123 Milan, Italy; lorenzo.drago@unimi.it; 3Department of Experimental Oncology, European Institute of Oncology IRCCS, 20141 Milan, Italy; sara.gandini@ieo.it (S.G.); federica.bellerba@ieo.it (F.B.)

**Keywords:** deep transcranial magnetic stimulation, obesity, gut microbiota, norepinephrine

## Abstract

Growing evidence highlights the crucial role of gut microbiota in affecting different aspects of obesity. Considering the ability of deep transcranial magnetic stimulation (dTMS) to modulate the cortical excitability, the reward system, and, indirectly, the autonomic nervous system (ANS), we hypothesized a potential role of dTMS in affecting the brain-gut communication pathways, and the gut microbiota composition in obesity. In a hospital setting, 22 subjects with obesity (5 M, 17 F; 44.9 ± 2.2 years; BMI 37.5 ± 1.0 kg/m^2^) were randomized into three groups receiving 15 sessions (3 per week for 5 weeks) of high frequency (HF), low frequency (LF) dTMS, or sham stimulation. Fecal samples were collected at baseline and after 5 weeks of treatment. Total bacterial DNA was extracted from fecal samples using the QIAamp DNA Stool Mini Kit (Qiagen, Italy) and analyzed by a metagenomics approach (Ion Torrent Personal Genome Machine). After 5 weeks, a significant weight loss was found in HF (HF: −4.1 ± 0.8%, LF: −1.9 ± 0.8%, sham: −1.3 ± 0.6%, *p* = 0.042) compared to LF and sham groups, associated with a decrease in norepinephrine compared to baseline (HF: −61.5 ± 15.2%, *p* < 0.01; LF: −31.8 ± 17.1%, *p* < 0.05; sham: −35.8 ± 21.0%, *p* > 0.05). Furthermore, an increase in *Faecalibacterium* (+154.3% vs. baseline, *p* < 0.05) and *Alistipes* (+153.4% vs. baseline, *p* < 0.05) genera, and a significant decrease in *Lactobacillus* (−77.1% vs. baseline, *p* < 0.05) were found in HF. *Faecalibacterium* variations were not significant compared to baseline in the other two groups (LF: +106.6%, sham: +27.6%; *p* > 0.05) as well as *Alistipes* (LF: −54.9%, sham: −15.1%; *p* > 0.05) and *Lactobacillus* (LF: −26.0%, sham: +228.3%; *p* > 0.05) variations. Norepinephrine change significantly correlated with *Bacteroides* (r^2^ = 0.734; *p* < 0.05), *Eubacterium* (r^2^ = 0.734; *p* < 0.05), and *Parasutterella* (r^2^ = 0.618; *p* < 0.05) abundance variations in HF. In conclusion, HF dTMS treatment revealed to be effective in modulating gut microbiota composition in subjects with obesity, reversing obesity-associated microbiota variations, and promoting bacterial species representative of healthy subjects with anti-inflammatory properties.

## 1. Introduction

Several environmental factors are involved in the pathophysiology of obesity: one emerging factor is the gut microbiota composition, which may influence both metabolism and eating behavior [1,2].

The relationship between gut microbiota and obesity is very complex and bidirectional. The role of a high-calorie diet in inducing changes in the gut microbiome composition and diversity is well known [3]. The balance between beneficial and detrimental gut microbiota could be disrupted, and the resulting impaired gut homeostasis can set the stage for several pathological conditions, as cancer [4]. The gut microbiota has been assumed to be a key link between obesity and cancer, via the generation of procarcinogenic toxic metabolites, metabolic dysregulation that contributes to tumor growth, and the induction of subclinical inflammation initiating tumorigenesis [5].

Gut microbiota composition in subjects with obesity differs significantly from healthy individuals for reduced gut microbiota richness, altered *Firmicutes* to *Bacteroidetes* ratio, and several changes in phyla, genera and species abundances [6].

One of the mechanisms through which gut microbiota may promote obesity is the increased energy harvest by specific microbes: the fermentation of indigestible carbohydrates by intestinal microbiota leads to increased intestinal absorption of monosaccharides and short-chain fatty acids (SCFAs), small organic metabolites produced by fermentation of dietary fibers [7]. SCFAs exert numerous physiologic effects in energy homeostasis, glucose regulation, and immune response. However, the role of SCFAs in obesity is not fully known: according to the type of ingested dietary fibers, SCFAs may contribute to or prevent obesity development [1]. Another mechanism through which microbiota may promote obesity is the increased gut permeability with translocation of bacteria or endotoxic bacterial components into active metabolic tissues, causing low-grade inflammation. SCFAs might indirectly affect metabolism through the release of gastrointestinal hormones [e.g., glucagon-like peptide-1 (GLP-1) and peptide YY)] influencing appetite neurons in the brain and inducing satiety [1,8]. The complex interplay between gastrointestinal hormones, microbiota-derived metabolites and energy homeostasis is identified as a part of the brain-gut axis [1]. The brain-gut signaling is bidirectional: the gut microbiota communicates with the brain via numerous metabolites, which are sensed by specialized gut’s cells, including entero-endocrine cells, entero-chromaffin cells, and primary or secondary afferent nerves [9]. The main metabolites involved in the brain-gut communication include: SCFAs (acetate, propionate and butyrate), gut-derived orexigenic (ghrelin, insulin) and anorexigenic signals (neuropeptide Y, GLP-1, cholecystokinin), stress mediators, inflammatory signals, and neuroactive compounds involved in the central nervous system (CNS), specifically in the “reward system” (e.g., serotonin, enkephalins, GABA, dopamine) [10] and in the autonomic nervous system (ASN) (e.g., biogenic amines: epinephrine, norepinephrine) [7].

On the other hand, the production of neuroactive compounds by gut microbiota could affect the host neurophysiology, influencing behavior in both human and animal model systems [11]. A large in situ production of biologically active neuroendocrine hormones by microbiota, such as dopamine and norepinephrine, was demonstrated in mice [12]. Furthermore, few human gut-derived strains of *Bifidobacterium dentium*, *Bifidobacterium infantis, Bifidobacterium adolescentis,* and *Lactobacillus brevis* were shown to produce gamma-amino-butyric acid (GABA) by using dietary monosodium glutamate [13,14]. For example, the administration of GABA-producing *Lactobacillus brevis* via soybean milk was revealed to be effective as an antidepressant in rats, such as fluoxetine [15]. Neuroactive compounds derived from gut microbiota metabolism act as potential mediators of communication between the gut microbiota and the host [16]. As evidence, some receptors and transporters critical for neuroactive compounds are found in enterocytes. For example, GABA transporters are present in the rat gastrointestinal tract [16]; dopamine receptors are widespread throughout the intestine [9,17].

In addition, the brain-gut communication pathways include ANS involvement: extrinsic afferent nerves are able to stimulate efferent nerve fibers, by directly or indirectly affecting local gut functions: motility, secretion, and thereby, microbiota composition [16]. Specifically, the ANS is closely connected with the enteric nervous system (ENS), a large and complex compartment of the peripheral nervous system, which integrates signals from the CNS through connections with the parasympathetic and sympathetic branches of the ANS, and sends information to CNS, via intrinsic primary afferent neurons projecting to motor neurons and interneurons [16].

An innovative technique that was proven to be effective in modulating the cortical excitability and the reward system is deep transcranial magnetic stimulation (dTMS). Deep TMS is based on the application of rapidly changing magnetic fields that are delivered with a special H-coil encased in helmet, placed over the scalp of the subject. Compared to traditional coil, the H-coil allows stimulation of deeper brain regions (3 vs. 1.5 cm from the skull), e.g., the insula, without generating significant currents in more superficial brain regions. The varying magnetic fields cause long lasting changes in cortical excitability and promote dopamine release [18,19,20]. For these properties, dTMS is widely used as a therapeutic tool in neurology and psychiatry. In addition to dopamine, repetitive dTMS induces modulation of other neurotransmitters involved in drug and food addiction-related processing (epinephrine, norepinephrine, GABA, and serotonin) [21,22], as well as in satiety/appetite hormones (leptin, ghrelin). Specifically, in subjects with obesity, a single high frequency dTMS session was able to increase β-endorphin levels, suggesting a dopaminergic reward system activation [23]. Recently, we demonstrated the efficacy [24] and safety [25] of a 5-week dTMS treatment, targeted bilaterally to the prefrontal cortex (PFC) and insula, in controlling food craving and reducing bodyweight, up to 1 year period in individuals with obesity, by enhancing inhibitory capacity of the PFC (specifically, medial orbitofrontal cortex) and thereby, improving control on eating behavior [26].

Although few and discordant studies have investigated the effects of dTMS on ANS, a recent meta-analysis showed that non-invasive brain stimulation is able to affect the ANS activity, especially when applied in the TMS shape and targeted to the PFC, instead of the primary motor cortex or other brain regions [27], suggesting reciprocal inter-relations between the ANS and brain areas, the central autonomic network (CAN) [28].

Although to date no clinical trials investigated the effects of TMS on gut microbiota composition in obesity, among the other neurostimulation techniques, transcranial Direct Current Stimulation (tDCS), targeted to the right dorsolateral PFC for 10 weeks, revealed effectiveness in inducing beneficial changes in the gut microbiome in an individual with overweight and cravings for sugary foods [29]. In addition, vagal nerve stimulation (VNS) exhibited a potential to modulate the enteric microbiota in preclinical studies [30].

Considering the ability of dTMS to modulate the cortical excitability, the reward system and the ANS, through the CAN, we hypothesized a potential role of dTMS in affecting the brain-gut communication pathways and thereby, the gut microbiota composition. In fact, an improvement of the gut microbiota composition might be considered one of the mechanisms by which the dTMS induces the already proven weight loss in individuals with obesity. Therefore, aims of the present study were to investigate the effects of a 5-week treatment with high frequency (HF), low frequency (LF) dTMS, or sham stimulation on gut microbiota composition in individuals with obesity, and to identify possible correlations between microbiota variations and metabolic, neurohormonal changes.

## 2. Results

### 2.1. Participants Characteristics

A total of 25 subjects with obesity were screened for the study; 22 patients (5 males and 17 females) met the study entry criteria and were enrolled in the study protocol. The mean age of the sample group was 44.9 ± 2.2 years, the mean weight was 104.3 ± 3.2 kg, and the mean BMI was 37.5 ± 1.0 kg/m^2^. At baseline, no significant differences in age and BMI were found among the three groups.

Patients fulfilling all inclusion/exclusion criteria were randomized into one of three experimental groups. Deep TMS stimulation conditions could either be HF (18 Hz group), LF (1 Hz group) or sham (sham group). Nine subjects with obesity were allocated in HF, six in LF, and seven in sham. At baseline, no significant differences in age, weight, BMI and measured parameters were found among the three groups (Table 1).

### 2.2. Bodyweight, Metabolic, and Neurohormonal Parameters

After 15 repetitive dTMS sessions, a significant weight loss was found in HF group compared to the other two groups (HF: −4.1 ± 0.8%, LF: −1.9 ± 0.8%, sham: −1.3 ± 0.6%, *p* = 0.042) (Figure 1a).

With respect to metabolism analysis, a significant decrease in resting energy expenditure (REE) percentage (−14.1 ± 4.3% vs. baseline, *p* = 0.009) and respiratory quotient (RQ) (−5.1 ± 1.9% vs. baseline, *p* = 0.033) was found in HF compared to baseline. No significant variations of REE and RQ were found in sham.

Following the first HF dTMS session, a significant increase in norepinephrine level was found (+28.6 ± 12.2% vs. baseline, *p* = 0.046). Norepinephrine did not acutely change in the other two groups (LF and sham). Chronically (after the 5-week treatment), norepinephrine significantly decreased both in HF (−61.5 ± 15.2% vs. baseline, *p* = 0.007) and LF (−31.8 ± 17.1% vs. baseline, *p* = 0.041) (Table 2 and Figure 1b).

### 2.3. Bacterial Changes at Phylum Level in Gut Microbiota Composition after 5-Week dTMS Treatment

After 5 weeks of HF dTMS, a reduction of *Bacteroidetes* (about −22.6%) and a negligible increase in *Firmicutes* (about +0.8%) were observed. In the LF group, *Bacteroidetes* decreased of about −20.8%, and *Firmicutes* of about −10.1%. As regard to the sham group, a mild decrease in both *Bacteroidetes* (about −3.2%) and *Firmicutes* (about −6.5%) was observed. Variation at phylum level was statistically significant only in the LF group.

### 2.4. Bacterial Changes at Genus Level in Gut Microbiota Composition after 5-Week dTMS Treatment

Significant differences at genus level detected in the gut microbiota of the three groups are shown in Figure 2A–C.

After 5 weeks of HF dTMS treatments, an increase in bacterial genera belonging to the *Bacteroidetes* phylum was found: *Alistipes* (reads abundance: 1490.7 ± 656.7 vs. 3777.8 ± 1137.8, +153.4%, *p* = 0.033 vs. baseline; *p* = 0.039 vs. sham; *p* = 0.029 vs. LF), and *Odoribacter* (reads abundance: 38.6 ± 20.0 vs. 261.7 ± 114.9, +577.2%, *p* = 0.059 vs. baseline; *p* = 0.024 vs. LF). Regarding the *Firmicutes* phylum, a decrease in the *Lactobacillus* genus (reads abundance: 260.1 ± 115.8 vs. 59.7 ± 33.1, −77.1%, *p* = 0.013 vs. baseline; *p* = 0.011 vs. sham) was observed; an opposite direction was highlighted for *Faecalibacterium,* which reads abundance significantly increased compared to baseline (3105.4 ± 1425.5 vs. 7897.3 ± 2314.3, +154.3%, *p* = 0.013); a trend to increase was observed for the *Clostridium* genus (reads abundance: 1720.0 ± 806.8 vs. 2547.2 ± 822.1, +48.1%, *p* = 0.058 vs. baseline; *p* = 0.088 vs. sham). In the *Proteobacteria* phylum, *Gemmiger* genus tended to increase after a 5-week HF treatment (reads abundance: 241.7 ± 116.9 vs. 422.0 ± 166.0, +74.6%, *p* = 0.076 vs. baseline).

No bacterial genus significantly changed after 5 weeks of LF dTMS treatment; in this group, only a significant average percentage reduction in the *Bacteroidetes* phylum was observed (−20.8% compared to baseline; *p* = 0.036).

In the sham group, as to the *Proteobacteria* phylum, significant increases of *Bilophila* (reads abundance: 896.8 ± 477.9 vs. 2856.1 ± 800.2, +218.5%, *p* = 0.036 vs. baseline; *p* = 0.013 vs. LF), and *Gemmiger* genera (reads abundance: 80.5 ± 41.6 vs. 1074.1 ± 494.4, +1234.9%, *p* = 0.036 vs. baseline; *p* = 0.008 vs. LF) were found.

All the variables analyzed (anthropometric measurements, biomarkers, and gut microbiota phyla and genera) at baseline (T0) and after 5 weeks of treatment (T2) in the three treatment groups have been shown in Appendix A.

### 2.5. Effects of 5-Week dTMS Treatment on Gut Microbiota Biodiversity

Shannon’s, Simpson’s, and Chao’s indices were calculated at T0 and T2 time-points to evaluate bacterial diversity and richness in the gut microbiota in the three groups (Figure 3A–C). No significant differences in gut microbiota biodiversity were observed after HF, LF dTMS, and sham treatment.

### 2.6. Correlations Between Variations in Metabolic/Neurohormonal Parameters and Gut Microbiota Composition

In the HF group, after 5 weeks of dTMS treatment, a significant correlation was found between the BMI variation and the increase in *Phascolarctobacterium* genus (r^2^ = 0.530; *p* = 0.026). Among neurohormonal parameters, norepinephrine decrease significantly correlated with several genera variations: *Eubacterium* (r^2^ = 0.734; *p* = 0.014), *Bacteroides* (r^2^ = 0.734; *p* = 0.014), and *Parasutterella* (r^2^ = 0.618; *p* = 0.036) (Figure 4). Details of the liner regression model have been shown in the Appendix A.

No significant correlations between bacterial genera abundance and norepinephrine variations were found both in LF and sham groups.

### 2.7. Adverse Events and Safety

No serious or severe side effects leading to the interruption of the treatment were observed. Individuals with obesity who received HF dTMS experienced more frequent headaches (5/9) than LF (2/6) and sham groups (2/7). This side effect resolved spontaneously within 3 days from the beginning of the treatment.

## 3. Discussion

Several preclinical and clinical studies suggest a relationship between gut microbiota composition and the pathogenesis of obesity. The ability of gut microbiota to produce and recognize neurochemical signals supports its potential role in promoting obesity, by influencing not only metabolism but also feeding behavior. In this study, taking into account the close relationship between brain and gut, and the sharing of common neurohormonal pathways, we hypothesized that dTMS treatment could induce weight loss in subjects with obesity also through indirect modulation of gut microbiota, besides as a result of a greater weight loss. The main finding of this study was an improvement of microbiota composition in subjects with obesity treated with a 5-week HF dTMS.

It is widely accepted that the gut microbiota in obesity is characterized by a lower level of complexity compared to that of healthy subjects [31]. The most common bacteria in human gut microbiota are members of the Gram-positive *Firmicutes* and the Gram-negative *Bacteroidetes* phyla; a minority of bacteria belongs to the *Proteobacteria* phylum. Obesity-associated gut microbiota is characterized by a reduced presence of species belonging to the *Bacteroidetes* phylum and a proportional greater abundance in the species of the *Firmicutes* phylum [32]. The altered *Firmicutes/Bacteroidetes* ratio results in the prevalence of species with an increased capacity to harvest energy from diet, and in a higher presence of enzymes for complex carbohydrate degradation and fermentation [33]. In our sample, the average of the *Firmicutes/Bacteroides* ratios highlighted a preponderance of the *Firmicutes* phylum, corroborating the notion that obesity is correlated with an imbalanced *Firmicutes/Bacteroides* ratio [34]. Relevant compositional differences in the gut microbiota of obese and lean individuals were revealed also at genera level. High concentrations of *Lactobacillus* genus were found in feces of individuals with obesity; specifically, a clear positive correlation exists between the abundance of *Lactobacillus reuteri* species and BMI [35]. Positive associations were also found between obesity and *Clostridium* cluster XIVa, *Escherichia coli* species, and *Staphylococcus* genus. Conversely, negative correlations were described between obesity and *Faecalibacterium prausnitzii*, *Clostridium* cluster IV, *Akkermansia muciniphila*, and *Methanobrevibacter smithii* species, *Bifidobacteria* and *Alistipes* genera; at the same time, the association with *Bacteroides* genus is controversial [36]. In our study, the improvement in the abundance of some previously described obesity-associated bacteria after 5 weeks of HF dTMS treatment, suggests that bacterial imbalance reported in our sample reflect the well-known obesity-associated changes in gut microbiota composition.

Furthermore, the findings of our study indicate that even just 5 weeks of repetitive HF dTMS treatment promote beneficial changes in microbial composition in subjects with obesity. These relevant changes in gut microbiota composition occurred in the same group in which a significant weight decrease has been concurrently observed, supporting our previous study [24]. Significant variations neither in bodyweight nor in the gut microbiota composition were observed in the groups of patients treated with LF dTMS or sham stimulation.

Specifically, after 5 weeks of HF dTMS treatment, we found an increase in bacterial genera belonging to the *Bacteroidetes* phylum (*Alistipes* and *Odoribacter*), suggesting a trend toward normalization of the *Firmicutes/Bacteroidetes* ratio. Previously, other studies showed *Bacteroidetes* increase during weight loss in subjects with obesity, suggesting that they may variate according to caloric intake change [34]. Among *Bacteroidetes*, *Alistipes* significantly increased in HF compared to other groups. *Alistipes* is considered a bacterial genus of particular interest in the field of obesity; a recent study testing the effects of a combined intervention (weight-loss program, exercise, behavioral therapy or bariatric surgery) in a population of subjects with obesity found significant changes in the *Alistipes* abundance during intervention [37,38]. Furthermore, a greater abundance of *Alistipes* was observed in participants successful in losing and maintaining their bodyweight over time [39]. *Odoribacter* is a butyric acid producing bacterial genus, also classified within the *Bacteroidetes* phylum. Interestingly, *Odoribacter* has been associated with a healthy fasting serum lipid profile [40], and showed a negative correlation with adiposity [41].

Within the *Firmicutes* phylum, a decrease in the *Lactobacillus* genus abundance was observed. The role of *Lactobacillus* in the gut microbiota composition of individuals with obesity is controversial: while some *Lactobacillus* species are associated with normal weight (*L. paracasei* or *L. plantarum*), other species (*L. reuteri*) present higher abundance in obesity [35,42]. Furthermore, the presence of *Lactobacillus* spp. in the obese gut microbiota has been positively associated with plasma high-sensitivity C-reactive protein [43], suggesting a pro-inflammatory role. The significant reduction of *Lactobacillus* genus observed in our sample after HF dTMS, could suggest a preponderance of species as *L. reuteri*. By increasing the sample size, future studies could be useful for exploring the relationships between selected species within the *Lactobacillus* genus, and their correlation with obesity. Within the *Firmicutes* phylum, an opposite direction has been highlighted for *Clostridium* genus, which abundance tends to increase in HF group. As for the *Lactobacillus*, its role is not univocal within the gut microbiota composition in obesity: according to the cluster, *Clostridium* positively correlates with bodyweight and BMI (e.g., *Clostridium histolyticum, Clostridium difficile*) [44,45], or is considered as a commensal of healthy gut (e.g., *Clostridium coccoides*, *Clostridium leptum*) [46]. The increase in *Clostridium* genus in our study could be explained also by a greater compliance with the prescribed Mediterranean diet in HF group; in fact, a study showed the ability of a Mediterranean-inspired anti-inflammatory diet to increase *Bacteroidetes* and *Clostridium* clusters and decrease in *Proteobacteria* and *Bacillaceae* population [47].

Interestingly, within the *Firmicutes* phylum, *Faecalibacterium* reads abundance significantly increased in HF compared to baseline. One of the mechanisms by which the gut microbiota affects obesity is the induction of systemic low-grade inflammation [48]. The gut microbiota increases mucosal permeability in obese mice, thereby promoting translocation of bacterial products (e.g., lipopolysaccharide) and stimulating the low-grade inflammation that is characteristic in obesity [48]. *Faecalibacterium* genus, especially *Faecalibacterium prausnitzii* species, has been recognized to have strong anti-inflammatory properties through butyrate production and induction of regulatory T cells and negatively correlates with inflammatory markers in subjects with obesity [49,50]. In our study, we found a significant increase in the abundance of *Faecalibacterium* after 5 weeks of HF dTMS treatment, supporting previous evidence of an increased abundance of bacterial species with anti-inflammatory properties, such as *Faecalibacterium prausnitzii*, in lean people as compared to subjects with obesity [42,51]. The beneficial impact of HF dTMS in *Faecalibacterium* genus within microbiota in obesity is comparable to that observed after bariatric surgery [52]. In our study, after a 5-week treatment, no significant differences in *Firmicutes/Bacteroidetes* ratio within the three groups were found, but in the two phyla, several bacterial genera significantly varied. The normalization of the *Firmicutes/Bacteroidetes* ratio could be likely counteracted by the significant beneficial increase in anti-inflammatory *Faecalibacterium* genus within the *Firmicutes* phylum.

Furthermore, in the HF group, a significant positive correlation was found between the *Phascolarctobacterium* genus increase and the BMI variation. *Phascolarctobacterium* genus (from family Veillonellaceae and order Clostridiales) is an SCFA producer. A higher abundance of *Phascolarctobacterium* has been shown in insulin-sensitive compared to insulin-resistant individuals [53]; a more recent study confirmed the positive association between *Phascolarctobacterium* and insulin sensitivity and highlighted a negative association with fasting insulin levels [54] and body fat mass. The increased abundance of *Phascolarctobacterium* is in line with a significant variation of fat mass observed in our sample after 5 weeks of HF dTMS, likely reflecting a concomitant improvement in insulin sensitivity.

The lack of significant differences in gut microbiota biodiversity observed in all three enrollment groups could be explained by the low sample size.

Finally, our findings indicate in the HF group a relationship between weight loss, rebalancing of the gut microbiota composition, and variation in neuro-hormone availability, specifically in norepinephrine levels. One of the main mechanisms, through which HF dTMS exerts its anti-obesity effects, is the modulation of food craving via an impact on the dopaminergic reward system. However, acute and chronic effects of repetitive TMS on the dopaminergic system should be distinguished. An important study in the history of TMS, measuring dopamine release in the striatum, following acute repetitive TMS on dorsolateral PFC (DLPFC), showed activation of cortico-striatal fibers leading to a focal dopamine release in the projection site of the stimulated cortical area [18]. With reference to chronic effects, the role of repetitive TMS in promoting dopamine release is controversial, as suggested by a neuroimaging study in which no measurable increase in the release of dopamine was detected after 10 daily sessions of HF repetitive TMS in humans [55]. Otherwise, a chronic modulatory effect of TMS on the dopamine transporter (DAT) availability has been identified [56]. In our study, similarly to dopamine, an acute increase in norepinephrine was found following the first HF dTMS session; otherwise, norepinephrine decreased after 5 weeks of HF dTMS treatment (Table 2). By studying the effects of chronic rTMS on gene expression of monoamine transporters in mice, a preclinical study highlighted the ability of rTMS to modulate monoamine transporter expression, increasing mRNA levels of DAT as well as of norepinephrine transporter (NET). In the latter case, an increased norepinephrine uptake and binding was also found in mouse brain, with a consequent reduced norepinephrine bioavailability [57]. In our study, we hypothesized that the described mechanisms could underlie the different behavior of the norepinephrine in response to acute or chronic stimulation, by reflecting the trend of dopamine. In fact, norepinephrine mainly derives from dopamine manipulation by dopamine B-hydroxylase, in peripheral tissue. In this study, we have reported a significant increase in norepinephrine after a single HF dTMS session, which was no longer present at the last stimulation session, and a significant reduction following a 5-week dTMS treatment (Table 2).

Gut norepinephrine availability could be mainly the result of a local (release from postganglionic sympathetic nerve fibers) or systemic (synthetization from chromaffin cells of the adrenal medulla) response. The systemic response is mainly controlled by the hypothalamus-pituitary-adrenal axis, while the local response is neural mediated, likely by the CAN (58). Different types of stressors, such as obesity, can increase not only local and plasma levels but also luminal levels of catecholamines as the norepinephrine in the gut, affecting proliferative activity of bacteria through different mechanisms: modulation of blood flow, nutrient absorption, gut motility, and interaction with the innate immune system [58]. For example, it has been reported that norepinephrine can stimulate proliferation of several enteric pathogens and increase the virulent properties of others as *Campylobacter jejuni* or *Escherichia coli* [59].

Our hypothesis underlying the correlation between norepinephrine decrease and microbiota composition improvement is summarized in Figure 5.

To demonstrate the beneficial effect of the decreased norepinephrine on the microbial balance in the gut, we found in the HF group a significant correlation between norepinephrine decrease and the following bacterial abundance variations: *Bacteroides, Eubacterium,* and *Parasutterella*. Concerning the *Bacteroides* genus, belonging to *Bacteroidetes* phylum, its abundance was found to decrease with an increase in BMI [60]; supporting previous evidence of a positive correlation with weight loss [44], also *Eubacterium* genus positively changed in our study; finally, *Parasutterella* was found to decrease in mice with diet-induced obesity and increase in controls [61], reflecting our finding of a milder increase in *Parasutterella* in HF group compared to other groups in association with a weight loss. Together these microbial variations, are suggestive of an ameliorating in gut microbiota composition, associated with the decrease in norepinephrine.

The present study has some limitations. This is an exploratory study, and no formal statistical hypotheses on the correlation between gut microbiome composition changes and norepinephrine variation were pre-specified to determine the sample size. Thus we did not adjust for multiple testing, and results will have to be validated in a further larger trial. However, considering the correlation between the norepinephrine and *Bacteroides* changes, a sample size of 23 achieves 80% power to detect a difference of 0.45 between the null hypothesis of low correlation (Spearman coefficient equal to 0.25) and the alternative hypothesis correlation of good correlation (Spearman = 0.7), using a two-sided hypothesis test with a significance level of 0.05. For the post-hoc power statistical calculation, we decided to correlate the main neurotransmitter used by the sympathetic nervous system and involved in the CAN, the norepinephrine, with the *Bacteroides* genus, which represents the most substantial portion of the mammalian gastrointestinal microbiota, where it plays a fundamental role in processing of complex molecules to simpler ones in the host intestine. Furthermore, *Bacteroides* genus belonging to *Bacteroidetes* phylum was found to be significantly associated with bodyweight changes. Therefore, we hypothesized that it was the bacterial genus that was most likely affected by dTMS treatment and most suggestive of an improvement in the gut microbiota composition.

The low number of enrolled individuals requires further and larger studies to confirm our results. Moreover, the sample was heterogeneous, and this could explain the variability in some hormonal values, e.g., leptin, norepinephrine, epinephrine.

In conclusion, only 5 weeks of HF dTMS treatment revealed to be effective in modulating gut microbiota composition in subjects with obesity, reversing obesity-associated microbiota variations, and promoting bacterial species representative of healthy subjects with anti-inflammatory properties. Gut microbiota rebalancing in obesity was found associated with a significant weight loss and a decrease in sympathetic activity, leading to hypothesize a specific effect of HF dTMS in improving gut microbiota composition through a modulatory action on sympathetic system. Modulation of gut microbiota composition through different types of intervention (restrictive diet, physical activity, dTMS) might represent a novel target to treat obesity.

## 4. Materials and Methods

### 4.1. Study Participants

This study was performed at the Endocrinology and Metabolic Diseases Division, IRCCS Policlinico San Donato, Italy.

Adult men and women (aged 22 to 65 years, inclusive), who were referred to the Endocrinology and Metabolic Diseases outpatient clinic for obesity treatment, were screened with a short interview to determine eligibility. If eligible, a more complete direct interview followed the telephone screening after providing signed informed consent. Inclusion criteria were a BMI ranging between 30 and 45 kg/m^2^ and the willingness to reduce bodyweight. Patients with a personal or a family history of seizures, as well as patients with psychotic disorders, organic brain disorders, any acute or chronic cardiovascular conditions, implanted metal devices, fasting blood glucose level > 150 mg/dl, abuse of substances other than nicotine, treatment with anti-obesity medications or medications associated with lowered seizure threshold were excluded from the study. Current medical history of chronic gut inflammatory diseases, gastrointestinal or any other type of cancer, short bowel syndrome, diarrhea or other symptoms of possible enteritis in the 14 days prior the screening visit, recent history of sigmoidoscopy or colonoscopy (within 14 days), current or recent (14 days prior the screening visit) use of antibiotics, prebiotics, probiotics, laxatives, antispasmodic, steroids, were also considered exclusion criteria.

### 4.2. Study Design

This study was a single-center, double-blind, sham-controlled, randomized clinical trial aimed at investigating the chronic effects of a 5-week repetitive dTMS treatment on the gut microbiota composition in subjects with obesity and to identify potential correlations between microbiota variations and bodyweight, metabolic, neurohormonal changes. Patients fulfilling all inclusion/exclusion criteria were randomized to one of three experimental groups: HF (18 Hz group), LF (1 Hz group), or sham (sham group). The range of stimulatory (18 Hz) and inhibitory (1 Hz) frequencies were based on previous literature evidence [24,62]. Patients were randomized in a 1:1:1 allocation ratio. Allocation in the three groups was performed according to a randomization sequence generated by a computerized program. The randomization code was only given to the treating investigator at the first treatment session by an independent investigator not involved with any other aspect of the trial. Participants and other investigators were unaware of the type of treatment assignment. The magnetic stimulation coil for active and sham treatments (dTMS sessions) was the same. Magnetic cards encoding for real or sham stimulation were used to activate the dTMS device or not, according to the randomization sequence. Both real and sham stimulation produced identical sounds and scalp sensations during the sessions.

Each patient received 15 treatment sessions [(3 times per week, in 5 weeks (visit 1–15)] of HF, LF dTMS, or sham stimulation. Patients were not administered any drugs or psychological or psychiatric therapy during the study period. Repetitive dTMS was the only treatment allowed.

### 4.3. Repetitive Deep Transcranial Stimulation Procedure (dTMS)

The repetitive dTMS was performed by a trained physician using a Magstim Rapid2TMS (The Magstim Co. Ltd., Whitland, Carmarthenshire, UK) stimulator equipped with an H-shaped coil, specifically designed to bilaterally stimulate the PFC and the insula [63,64]. Magnetic cards encoding for real or sham stimulation were used to activate the dTMS device. Both real and sham stimulation produced identical sounds and scalp sensations during the sessions.

The characteristics of the stimulation protocols are the same as those used in the study by Ferrulli et al. [24]. Details of the stimulation procedure have been reported in the Appendix A.

High-frequency sessions consisted of 80 trains of 18 Hz, each lasting 2 s, with an inter-train interval of 20 s. The HF treatment duration was 29.3 min with 2880 pulses in total. Low-frequency sessions consisted of four trains of 1 Hz, each lasting 10 min, with an inter-train interval of 1 min. The LF treatment duration was 43 min with 2400 pulses in total. The sham treatment was performed by a sham coil located in the same case as the real coil, producing similar acoustic artifacts and scalp sensations, inducing only negligible electric fields in the brain. In all groups receiving the real treatment, the stimulation was performed with an intensity of 120% of the RMT.

### 4.4. Diet and Lifestyle Recommendations

During the entire study, all subjects were prescribed a hypocaloric diet. The energy requirement was calculated by the dietitian based on the measured basal metabolic rate (via indirect calorimetry) and the physical activity of each subject identified at the screening visit. A total of 300 kcal/day were subtracted from this amount of energy to obtain the recommended hypocaloric diet. The daily dietary intake included approximately 45% to 50% calorie intake from carbohydrate, up to 30% of calories from fat, and 20% to 25% of calories from protein. The total carbohydrate amount included about 20–25 gr/day of fibers. According to the Italian Society of Human Nutrition (SINU) guidelines, the fibers amount has been computed as 16 gr/1000 Kcal/day. At each follow-up visit, the dietitian confirmed the reduction of food intake with a direct interview. The use of probiotics and yogurt was not allowed during the study period.

### 4.5. Measurements

#### 4.5.1. Anthropometric Values

Anthropometric measurements were recorded at baseline at the last dTMS session (visit 15). They included: bodyweight and height to calculate BMI (kg/m^2^).

#### 4.5.2. Body Composition

The BOD POD (BOD POD^®^ Body Composition System, COSMED, Agrate Brianza Italy), which uses air displacement plethysmography, was employed to determine body composition and, specifically, fat mass percentage. The BOD POD showed to be both valid and reliable for body composition determination [65]. The BOD POD was shown to be as valid as dual-energy X-ray absorptiometry [66]. Fat mass (FM) and fat-free mass (FFM) percentages were considered as body composition parameters. Body composition was evaluated at baseline visit and at visit 15.

#### 4.5.3. Resting Energy Expenditure (REE) and Respiratory Quotient (RQ)

Metabolism analysis was performed by measuring the REE and the RQ. After an overnight fast, REE was measured by indirect calorimetry, using an open-circuit calorimeter (Sensor Medics, Milan, Italy). Indirect calorimetry is the reference method for energy expenditure determination [67,68]. The REE was assessed continuously during the indirect calorimetry procedure, with the subjects lying supine without talking nor sleeping for 30 min at a room temperature ranging between 22 and 23 °C. The REE measure for each participant considers the mean of the last 25 min of the analysis. The RQ was calculated as the ratio between fluxes of released CO_2_ and consumed O_2_ derived from the oxidation of the substrates. The value of the RQ depends on what type of substrate (glucose, lipids, or proteins) is being oxidized. Indirect calorimetry was performed at baseline visit and at visit 15.

#### 4.5.4. Laboratory Measurements

Blood tests were carried out before the first dTMS session (T0) and immediately afterward (T1), before the last dTMS session (T2), and immediately afterward (T3). After a 12 h overnight fast, a Venflon catheter was placed into an antecubital vein of each participant to draw blood. Blood samples were centrifuged for 15 min at 2000× *g*. A part of the blood was immediately processed; about 10 mL of every sample was stored in aliquots at −80 °C. Metabolic and neurohormonal determinations were performed by standardized techniques.

The metabolite assessment included: glucose (mg/dL), fructosamine (µmol/L), glycated hemoglobin (mmol/mol), cholesterol (mg/dL), triglycerides (mg/dL).

The neurohormonal assessment included: insulin (µU/mL), glucagon (pg/mL), leptin (ng/mL), ghrelin (ng/mL), β-endorphins (ng/mL), epinephrine (pg/mL), norepinephrine (ng/mL), TSH (µUI/mL) and salivary cortisol (µg/dL).

#### 4.5.5. Gut Microbiota Analysis

Fecal samples were collected in sterile containers before and after 15 dTMS sessions and were stored at −20 °C until analysis.

##### DNA Extraction

Total DNA was extracted from fecal samples using the QIAamp DNA Stool Mini Kit following the manufacturer’s instructions (Qiagen, Milan, Italy).

##### 16 S rRNA Gene Amplification

Partial 16S rRNA gene sequences were amplified from extracted DNA using the 16S Metagenomics Kit (Life Technologies, Monza, Italy) that is designed for rapid analysis of polybacterial samples using Ion Torrent sequencing technology. The kit includes two primer sets that selectively amplify the corresponding hypervariable regions of the 16S region in bacteria: primer set V2-4-8 and primer set V3-6, 7-9. The PCR conditions used were 10 min at 95 °C, 30 cycles of 30 s at 95 °C, 30 s at 58 °C, and 20 s at 72 °C, followed by 7 min at 72 °C. Amplification was carried out by using a SimpliAmp thermal cycler (Life Technologies, Monza, Italy). The integrity of the PCR amplicons was analyzed by electrophoresis on 2% agarose gel.

##### Ion Torrent PGM Sequencing of 16S rRNA Gene-Based Amplicons

The PCR products derived from amplification of specific 16S rRNA gene hypervariable regions were purified by a purification step involving the Agencourt AMPure XP DNA purification beads (Beckman Coulter Genomics, Krefeld, Germany) in order to remove primer dimers. DNA concentration of the amplified sequence library was estimated through the Qubit system (Life Technologies, Monza, Italy). From the concentration and the average size of each amplicon, the number of DNA fragments per microliter was calculated, and libraries were created by using the Ion Plus Fragment Library kit (Life Technologies, Monza, Italy). Barcodes were also added to each sample using the Ion Xpress Barcode Adapters 1–16 kit (Life Technologies, Monza, Italy). Emulsion PCR was carried out using the Ion OneTouch TM 400 Template Kit (Life Technologies, Monza, Italy) in conformity with the manufacturer’s instructions. Sequencing of the amplicon libraries was carried out on a 316 chip using the Ion Torrent Personal Genome Machine (PGM) system and employing the Ion PGM Hi-Q kit (Life Technologies, Monza, Italy) according to the supplier’s instructions. After sequencing, the individual sequence reads were filtered by the PGM software to remove low-quality and polyclonal sequences. Sequences matching the PGM 3′ adaptor were also automatically trimmed. A total of 16 rRNA sequences were then analyzed by Ion Reporter Software, which comprises a suite of bioinformatics tools that streamline and simplify the analysis of semiconductor-based sequencing data. The 16S rRNA workflow module in Ion Reporter Software was able to classify individual reads combining a Basic Local Alignment Search Tool (BLAST) alignment to the curated Greengenes database, which contains more than 400,000 records, with a BLAST alignment to the premium curated MicroSEQ ID database, a high-quality library of full-length 16S rRNA sequences. In the first step, reads were aligned to the MicroSEQ ID library with any unaligned reads subject to a second alignment to the Greengenes database to achieve rapid and exhaustive bacterial identification. The final output of Ion Reporter Software was the identification and abundance of microorganisms at phyla, class, family, and genus levels.

#### 4.5.6. Statistical Analysis

Data of each parameter were expressed as mean ± standard error of the mean (SEM), and percent variations between final and baseline values were indicated. One-way ANOVA followed by Bonferroni post-hoc test was performed for normally distributed values. Prior to conducting the ANOVA, the assumption of normality was evaluated and determined with the Shapiro–Wilk test. Non-normally distributed values were studied with the Kruskal–Wallis test. Comparisons between the matched groups were obtained with the paired Student’s *t*-tests for normally distributed values and with the Wilcoxon signed-rank test for non-normally distributed data. Spearman correlation test and linear regression models were used to find correlations and relationships among parameters.

For all tests, differences were considered statistically significant at *p* ≤ 0.05.

Power calculation. The aim of the study was to investigate the effects of a 5-week treatment on gut microbiota composition, looking at the correlations between microbiota variations and metabolic, neurohormonal changes. In particular, considering the correlation between the norepinephrine change and *Bacteroides*, a sample size of 23 achieves 80% power to detect a difference of 0.45 between the null hypothesis correlation of 0.25 and the alternative hypothesis correlation of 0.70 using a two-sided hypothesis test with a significance level of 0.05.

Analyses were carried out with the Statistical Package SPSS version 24 for Windows (IBM Corp.; Armonk, NY, USA), Excel 2010 (Microsoft; Redmond, WA, USA), and XLSTAT 2014 (Addinsoft; New York, NY, USA).

## Figures and Tables

**Figure 1 ijms-22-04692-f001:**
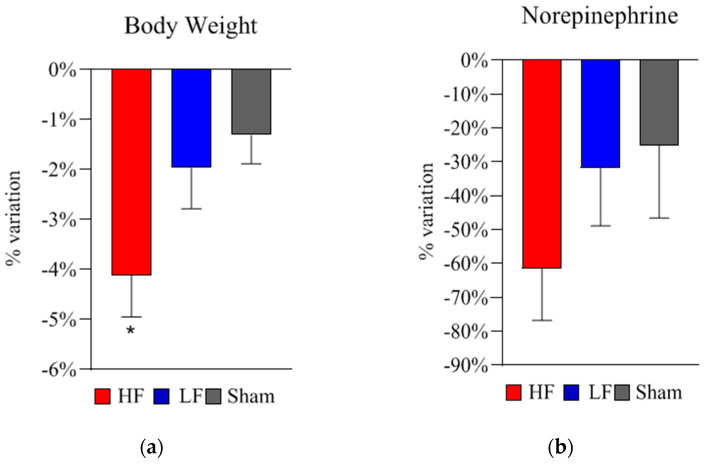
Deep TMS treatment effects on bodyweight and norepinephrine levels in the three groups. (**a**) Treatment effects on bodyweight. Bodyweight variation between baseline and following the 5-week treatment is shown for all groups (HF, high frequency 18 Hz; LF, low frequency 1 Hz; sham). The panel presents the average percent variations (mean% ± SEM) in bodyweight after the 5-week treatment (HF: −4.1 ± 0.8%, LF: −1.9 ± 0.8%, sham: −1.3 ± 0.6%, * *p* = 0.042). Analysis revealed a significant decrease in bodyweight in HF compared to the other two groups (* *p* = 0.042). (**b**) Treatment effects on norepinephrine. Norepinephrine variation between baseline and following the 5-week treatment is shown for all groups (HF, high frequency 18 Hz; LF, low frequency 1 Hz; sham). The panel presents the average percent variation (mean% ± SEM) in norepinephrine levels compared to baseline. Analysis revealed a significant decrease in norepinephrine in HF (−61.5 ± 15.2% vs. baseline, *p* = 0.007) and, although smaller, in LF (−31.8 ± 17.1% vs. baseline, *p* = 0.041). Comparing norepinephrine variation percentages between groups, no significant differences were found. Statistical analysis was performed using the two-tailed t-test. Legend: *SEM =* standard error of the mean; ** p <* 0.05.

**Figure 2 ijms-22-04692-f002:**
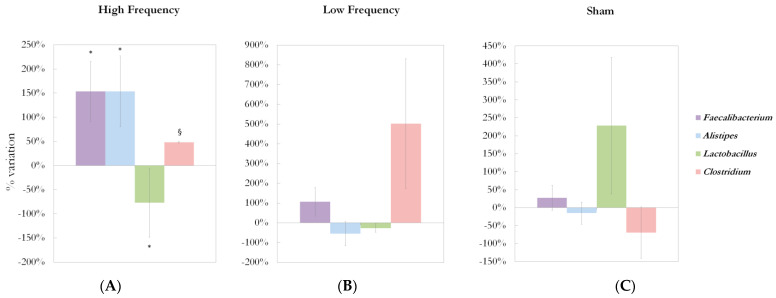
Deep TMS effects on gut microbiota composition in the three treatment groups. (**A**) Treatment effects on *Faecalibacterium, Alistipes, Lactobacillus, Clostridium* genera abundances in HF group. *Faecalibacterium, Alistipes, Lactobacillus, Clostridium* abundance percent change from baseline to 5-week treatment are shown (mean% ± SEM). Analysis revealed a significant increase in *Faecalibacterium* (reads abundance: 3105.4 ± 1425.5 vs. 7897.3 ± 2314.3, +154.3%, * *p* = 0.013) and *Alistipes* (reads abundance: 1490.7 ± 656.7 vs. 3777.8 ± 1137.8, +153.4%, * *p* = 0.033 vs. baseline; * *p* = 0.039 vs. sham; * *p* = 0.029 vs. LF), a significant decrease in *Lactobacillus* (reads abundance: 260.1 ± 115.8 vs. 59.7 ± 33.1, −77.1%, * *p* = 0.013 vs. baseline; * *p* = 0.011 vs. sham), and a trend to increase in *Clostridium* (reads abundance: 1720.0 ± 806.8 vs. 2547.2 ± 822.1, +48.1%, § *p* = 0.058 vs. baseline; § *p* = 0.088 vs. sham) after 5 weeks of HF dTMS treatment. (**B**) Treatment effects *on Faecalibacterium, Alistipes, Lactobacillus, Clostridium* genera abundance in LF group. *Faecalibacterium, Alistipes, Lactobacillus, Clostridium* abundance percent variation between baseline and after the 5-week treatment is shown for low frequency (LF, 1 Hz) group. No significant variations were found in *Faecalibacterium, Alistipes, Lactobacillus, Clostridium* abundance in the LF group. (**C**) Treatment effects on *Faecalibacterium, Alistipes, Lactobacillus, Clostridium* genera abundance in the sham group. *Faecalibacterium, Alistipes, Lactobacillus, Clostridium* abundance percent variation between baseline and after the 5-week treatment is shown for the sham group. No significant variations were found in *Faecalibacterium, Alistipes, Lactobacillus, Clostridium* abundance in the sham group. Statistical analysis was performed using the two-tailed t-test. *Legend: SEM = standard error of the mean; * p <* 0.05; § 0.05 *≤ p <* 0.1.

**Figure 3 ijms-22-04692-f003:**
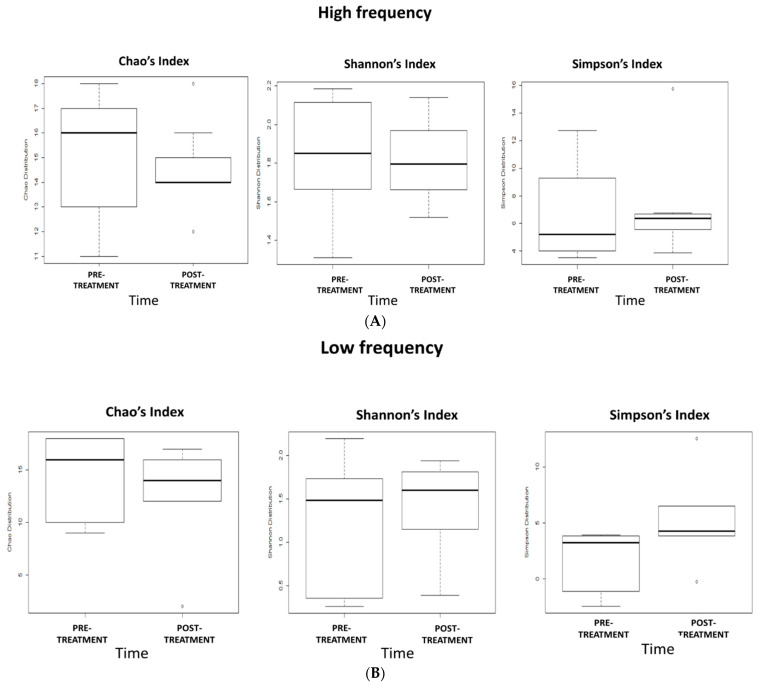
Effects of 5-week dTMS treatment on gut microbiota biodiversity evaluated by Chao’s, Shannon’s, and Simpson’s indices in the 3 treatment groups: (**A**) HF, (**B**) LF, (**C**) sham.

**Figure 4 ijms-22-04692-f004:**
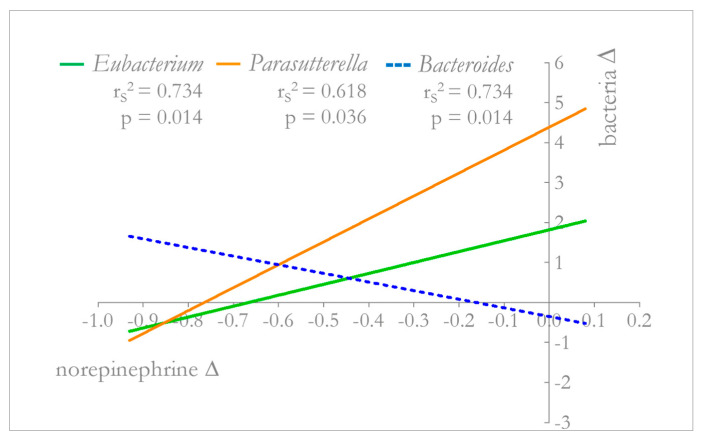
Correlation between norepinephrine and *Eubacterium, Parasutterella, Bacteroides* abundance variations in HF group. In the HF group, after 5 weeks of dTMS treatment, a significant correlation was found between norepinephrine decrease and several genera variations: *Eubacterium* (r^2^ = 0.734; *p* = 0.014), *Parasutterella* (r^2^ = 0.618; *p* = 0.036), and *Bacteroides* (r^2^ = 0.734; *p* = 0.014). Spearman correlation test and linear regression models were used for finding correlations and relationships among parameters.

**Figure 5 ijms-22-04692-f005:**
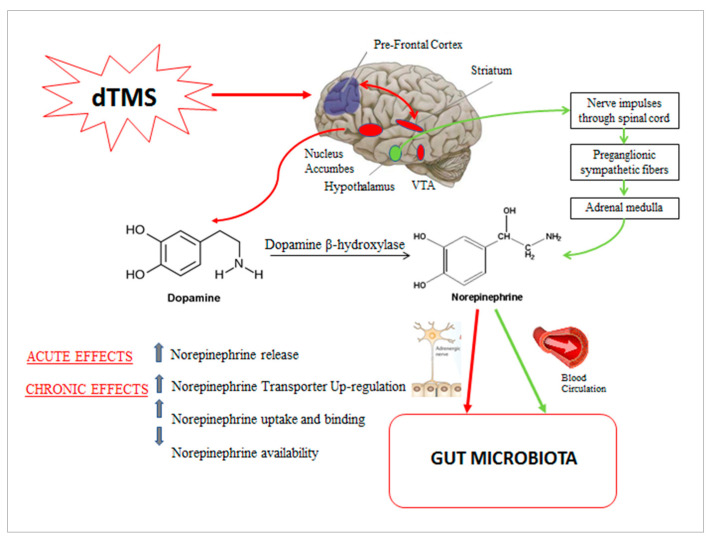
Possible mechanisms involved in the relationship between dTMS, rebalancing of the gut microbiota composition and norepinephrine variation. Deep TMS by stimulating the PFC at high frequency could acutely induce activation of the dopaminergic reward system (striatum, VTA, nucleus accumbens), promoting dopamine release. Norepinehrine mainly derives from dopamine manipulation by dopamine B-hydroxylase; norepinephrine could be locally released in the gut through postganglionic sympathetic nerve fibers. At the same time, dTMS could induce a HPA axis-mediated systemic response, promoting norepinephrine synthetization from chromaffin cells of the adrenal medulla; norepinephrine achieves the gut through the bloodstream. Chronic treatment with dTMS has been suggested to induce a NET modulation, with a consequent central decrease in norepinephrine outflow. We hypothesized that the decreased systemic norepinephrine results in a reduced gut luminal norepinephrine, with consequent beneficial effects on gut microbiota composition. Legend: dTMS = deep transcranial magnetic stimulation; PFC= prefrontal cortex; VTA= ventral tegmental area; HPA = hypothalamic-pituitary-adrenal; NET = norepinephrine transporter.

**Table 1 ijms-22-04692-t001:** Baseline characteristics of participants *.

	Unit of Measure	HF (*n* = 9)	LF (*n* = 6)	Sham (*n* = 7)	*p*-Value
Age, years	Years	44.2 ± 3.5	41.8 ± 4.5	48.4 ± 3.9	0.453
Sex	Males/females	3/6	1/5	1/6	-
BMI	kg/m^2^	37.3 ± 2.0	40.7 ± 2.1	35.5 ± 1.0	0.453
FM	%	46.4 ± 2.1	49.6 ± 2.5	46.3 ± 2.0	0.711
REE	%	101.0 ± 3.0	94.0 ± 6.2	93.1 ± 5.0	0.189
RQ		0.89 ± 0.02	0.89 ± 0.02	0.85 ± 0.03	0.144
Glucose	mg/dL	95.0 ± 5.9	103.2 ± 14.9	95.9 ± 6.5	0.940
Insulin	µU/mL	27.1 ± 8.1	21.6 ± 2.6	20.1 ± 6.7	0.740
HOMA-IR	-	7.2 ± 2.5	5.6 ± 1.1	4.8 ± 1.7	0.798
Glucagon	pg/mL	40.7 ± 3.7	34.4 ± 2.2	43.2 ± 4.9	0.072
Cholesterol	mg/dL	208.9 ± 11.3	179.0 ± 16.4	189.0 ± 7.2	0.995
Triglycerides	mg/dL	146.1 ± 24.7	143.7 ± 35.1	103.1 ± 17.1	0.836
Fructosamine	µmol/L	243.9 ± 10.0	221.7 ± 8.8	234.1 ± 14.1	0.842
Glycated hemoglobin	mmol/mol	36.9 ± 1.5	41.0 ± 7.1	34.8 ± 2.8	0.235
TSH	µUI/mL	2.4 ± 0.3	2.38 ± 0.7	3.2 ± 0.7	0.354
Cortisol	µg/dL	0.43 ± 0.06	0.40 ± 0.06	0.35 ± 0.05	0.151
Ghrelin	ng/mL	13.1 ± 3.5	13.0 ± 3.6	7.9 ± 2.9	0.316
Leptin	ng/mL	64.7 ± 13.1	83.4 ± 22.5	108.3 ± 49.9	0.642
Epinephrine	pg/mL	1356.0 ± 261.8	996.7 ± 151.1	1487.7 ± 446.2	0.642
Norepinephrine	ng/mL	4.7 ± 0.8	4.7 ± 0.9	4.2 ± 0.9	0.400
β-endorphin	ng/mL	0.708 ± 0.12	0.742 ± 0.21	0.549 ± 0.09	0.878

* Data are shown as mean ± standard error of the mean (SEM) except for sex. Comparisons among groups were obtained with ANOVA followed by Bonferroni post-hoc test for normally distributed values and with Kruskal–Wallis for non-normally distributed values. Abbreviations: HF = high frequency; LF = low frequency; BMI = body mass index; FM = fat mass; REE= resting energy expenditure; RQ = respiratory quotient; HOMA-IR= homeostatic model assessment-insulin resistance; TSH = thyroid-stimulating hormone.

**Table 2 ijms-22-04692-t002:** Acute and chronic variations of norepinephrine levels. Data are shown as mean ± standard error of the mean (SEM). Average values of norepinephrine ± SEM are reported at baseline (T0), after the first dTMS session (T1), before (T2), and after the last (15th) dTMS session (T3) for all three groups (HF, high frequency 18 Hz; LF, low frequency 1 Hz; sham). Comparisons at different times (T0 vs. T1, T2 vs. T3, T0 vs. T2) were obtained with two-tailed Student’s *t*-tests. For all tests, differences were considered statistically significant at *p* ≤ 0.05. Legend: HF = high frequency; LF = low frequency; * *p* < 0.05; ** *p* < 0.01.

	Norepinephrine (ng/mL)						
	T0	T1	T0 vs. T1	T2	T3	T2 vs. T3	T0 vs. T2
HF Group	4.7 ± 0.8	5.7 ± 0.7	* 0.046	1.8 ± 0.7	1.7 ± 0.6	0.250	** 0.007
LF Group	4.7 ± 0.9	4.0 ± 1.2	0.261	3.1 ± 0.7	2.2 ± 0.4	0.259	* 0.041
Sham Group	4.2 ± 0.9	3.6 ± 1.4	0.465	3.4 ± 1.5	4.3 ± 1.6	0.578	0.473

## Data Availability

Individual participant data that underlie the results reported in this article, after deidentification (text, tables, figures, and appendices), together with the study protocol, will be available, beginning 9 months and ending 36 months following article publication. Data will be available for investigators whose proposed use of the data has been approved by an independent review committee (learned intermediary) identified for this purpose and for individual participant data meta-analysis.

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
