# Peer review of "Deep Transcranial Magnetic Stimulation Affects Gut Microbiota Composition in Obesity: Results of Randomized Clinical Trial"

_ijms, 2021, doi:10.3390/ijms22094692_

Round 1
Reviewer 1 Report
Thanks for the opportunity of review this interesting and novel study.
My major concerns included that the sample size is very small and the allocation of participants into three groups was not clearly described. Also, the primary outcome of this study is not very clear for me. Therefore, if a less common design is employed here, please explain the choice. Especially I feel: the authors study design may imply the need for a larger sample size or more complex analysis and interpretation.
More details please see below:
Title:
Please indicate the study design. Is this study “randomised”?
Abstract:
Line 19: Were participants randomly allocated into 3 groups? Please indicate
Please also indicate where the data were collected? In a hospital setting?
For the primary outcome, please report a result for each group and the estimated effect size and its precision
Introduction:
Is there a reference to a systematic review of previous similar trials that indicated the need for this new study? Please justify in the introduction. Or please to note of the absence of such trials.
Methods:
4.2. Study design: again, please indicate were participants randomly allocated into three groups. If done, who were blinded?
4.3. Line 560: Please briefly describe ref 19 and the reasons of using this protocol.
4.5.6. Please add the power calculation; how the sample size were determined? Please add more details to the liner regression model
Results:
Table 1: please add the results of after “intervention”
Perhaps, Fig 5 move to the Discussion?
Discussion:
Please add Limitation and Strength of your study.
Author Response
Response to Reviewer 1
We would like to thank the reviewer for his/her interest in our work and his/her constructive comments aimed at improving the quality of our paper. Below the point to point answers/clarifications required:
Please see the attachment
- Title: Please indicate the study design. Is this study “randomised”?
Study title has been changed according to the Reviewer suggestion: a reference to the study design has been added. The new title is: “Deep Transcranial Magnetic Stimulation affects gut microbiota composition in obesity: results of randomized clinical trial”
- Abstract: Line 19: Were participants randomly allocated into 3 groups? Please indicate. Please also indicate where the data were collected? In a hospital setting?
A reference to the study design and setting has been added also in the abstract; a more detailed description has been added in the Materials and Method section.
- Abstract: For the primary outcome, please report a result for each group and the estimated effect size and its precision
As rightly suggested by the Reviewer 1, the effects of the treatment on the primary outcome (variation of the Genera or Phyla abundances in the gut microbiota of obese individuals, with increase of phyla/genera with higher anti-inflammatory properties) for the other 2 groups (LF and Sham) have been reported in the abstract. Furthermore, for better understanding, we added a table (Table S1 in the Supplementary Material) in which we reported, in more detail, the values of all the examined variables, including phyla and genera abundance, before and after the intervention for each treatment group. This is an exploratory study and no study previously investigated the effects of TMS on gut microbiota composition, although the hypothesis that a neurostimulation treatment could modulate the gut microbiota by influencing different neurotransmitter systems is supported by the numerous evidence of a brain-gut-microbiota axis.
- Introduction: Is there a reference to a systematic review of previous similar trials that indicated the need for this new study? Please justify in the introduction. Or please to note of the absence of such trials.
Nowadays, no clinical trials investigated the effects of TMS on gut microbiota composition not only in individuals with obesity but also in other population groups. However, among the other neurostimulation techniques, only a case of an individual with overweight and cravings for sugary foods, treated with anodal transcranial Direct Current Stimulation (tDCS) has been reported. In this case, a 10-week treatment with tDCS, addressed to the right dorsolateral prefrontal cortex, has been found to induce beneficial changes in the intestinal microbiome (Artifon M, Schestatsky P, Griebler N, Tossi GM, Beraldo LM, Pietta-Dias C. Effects of transcranial direct current stimulation on the gut microbiome: A case report. Brain Stimul. 2020 Sep-Oct;13(5):1451-1452. doi: 10.1016/j.brs.2020.07.019). Also vagal nerve stimulation (VNS), a commonly used technique for multiple disorders, exhibits a potential to modulate the enteric microbiota, enabling investigation and possibly treatment of numerous neurologic disorders in which the microbiota has been linked with disease (Haney MM, Ericsson AC, Lever TE. Effects of Intraoperative Vagal Nerve Stimulation on the Gastrointestinal Microbiome in a Mouse Model of Amyotrophic Lateral Sclerosis. Comp Med. 2018;68(6):452-460. doi:10.30802/AALAS-CM-18-000039).
Introduction section has been completed with these specifications.
- Methods:2. Study design: again, please indicate were participants randomly allocated into three groups. If done, who were blinded?
The study was designed as a double-blind, sham-controlled, randomized clinical trial. A detailed explanation about the randomization, masking and blinding procedures has been added in the Materials and Methods section (4.2 Study Design).
- Line 560: Please briefly describe ref 19 and the reasons of using this protocol.
This exploratory study is part of a larger randomized, double-blinded, placebo-controlled clinical trial designed to study the effects of the dTMS treatment aimed at reducing body weight and food craving (Ferrulli A, Macrì C, Terruzzi I, Massarini S, Ambrogi F, Adamo M, Milani V, Luzi L. Weight loss induced by deep transcranial magnetic stimulation in obesity: A randomized, double-blind, sham-controlled study. Diabetes Obes Metab. 2019 Aug;21(8):1849-1860). This project has been registered with ClinicalTrials.gov, number NCT03009695. After demonstrating the effectiveness of dTMS in promoting weight loss in the main study, we employed the same stimulation protocols for all sub-studies of the project. Details of stimulation protocols have been entered in the Supplementary Materials.
- 5.6. Please add the power calculation; how the sample size were determined?
A section on the power calculation has been added in Materials and Methods.
- Please add more details to the liner regression model
As suggested by the Reviewer, details about the linear regression model have been defined and entered in a table in the Supplementary Material (Table S2).
- Results: Table 1: please add the results of after “intervention”
A table reporting the values of all the examined variables, before and after the intervention for each treatment group, has been added in the Supplementary Material (Table S1)
- Perhaps, Fig 5 move to the Discussion?
As suggested by the Reviewer, Figure 5 has been moved in the Discussion section.
- Discussion: Please add Limitation and Strength of your study.
A paragraph on the limitations and strengths of your study has been added in the Discussion section.

Reviewer 2 Report
The study from Ferrulli and colleagues investigate the use of deep transcranial magnetic stimulation on the gut microbiota composition in obese patients. The manuscript is well written, and the overall experimental design was well executed. My main concern is how robust and reproducible is the findings of this study. There were only 22 subjects divided in 3 experimental groups. The authors did not provide evidence that a type II error has not occurred. I suggest the authors run statistical power analysis to confirm that the correct number of samples has been used. Indeed, the authors had suggested in the discussion that the lower “n” could have impacted some of their results.
Other comments:
L38 on: Is the difference found in the gut microbiota composition in obese due to the most likely unhealthy diet of these people or the reason for them to be obese?
L53: the role of GI hormones is not referenced.
L69-70: Can the production of GABA by L. Brevis and B. dentium be reproduced in live organism? Can we say that these strains are the most efficient GABA-producer even if the original paper is just comparing 91 strains?
L124: How were the experimental groups assigned?
L536: just to be clear, was alcohol intake allowed? Since nicotine was allowed, were all forms (smoked, skin absorption…) accepted?
L542: how did the authors control the use of prebiotics naturally occurring on the diet (eg. Granola, oat, other fibres…)?
L618: how the fecal samples were collected and how long it took to be frozen?
Table 1: Leptin and Epinephrine have a higher SEM compared to the other two groups, and specially the leptin Sham group has a mean that is almost the double that HF. Could you explain why of this variability? Are these and other data normal? Was performed a power analysis to identify the correct number of subjects per group? Why HF group has 9 subjects while LF and Sham have 6 and 7, respectively?
L146-152: Where are these data in the manuscript? Were remaining phyla not statistically significant or not assessed?
Figure 2 legend and description: I found a bit difficult to follow the statistical analysis on figure 2. It is not clear when the data refers to comparison within treatment or before-after treatment.
Figure 3 needs to be redesigned.
Author Response
Response to Reviewer 2
We really appreciated the Reviewer's interest toward our paper and his/her constructive comments aimed at improving the quality of our paper. We are aware that a limit of the study is the small sample size, however as we stated in the discussion section, this is an exploratory study and no formal statistical hypotheses on the correlation between gut microbiome composition changes and other biomarker variations were pre-specified to determine the sample size. However, as rightly suggested by the Reviewer, we performed the statistical power analysis, considering the significant correlation between the norepinephrine and Bacteroides changes, and the results confirmed the correct sample size (see Discussion section).
Below the point to point answers/clarifications required:
- L38 on: Is the difference found in the gut microbiota composition in obese due to the most likely unhealthy diet of these people or the reason for them to be obese?
The relationship between gut microbiota and metabolic diseases, specifically obesity, is very complex and bidirectional. The role of a high-calorie diet in inducing changes in the gut microbiome composition and function is by now approved by the scientific community (Guirro et al. Effects from diet-induced gut microbiota dysbiosis and obesity can be ameliorated by fecal microbiota transplantation: a multiomics approach. PLoS One. 2019;14:e0218143. doi: 10.1371/journal.pone.0218143). On the other hand, it has been suggested that obesity may also result from earlier perturbations of the gut microbiome, which affect metabolic function and energy homeostasis. Nutritional, lifestyle and genetic factors could be involved in an altered microbiota composition in obesity (Muscogiuri et al. Gut microbiota: a new path to treat obesity. Int J Obes Suppl. 2019 Apr;9(1):10-19. doi: 10.1038/s41367-019-0011-7). Therefore, this question cannot be answered with a univocal answer.
- L53: the role of GI hormones is not referenced.
As suggested by the Reviewer, two references for the role of GI hormones have been entered.
- L69-70: Can the production of GABA by L. Brevis and B. dentium be reproduced in live organism? Can we say that these strains are the most efficient GABA-producer even if the original paper is just comparing 91 strains?
GABA-producing Lactobacilli are mainly isolated from food products such as cheese, yogurt, sourdough…The ability to produce GABA by human-derived Lactobacilli and Bifidobacteria remains poorly studied. In fact, only few gut-derived strains of B. dentium, B. infantis, B. adolescentis and L. brevis were shown to produce GABA (Barrett et al. γ-Aminobutyric acid production by culturable bacteria from the human intestine. J Appl Microbiol. 2012 Aug;113(2):411-7. doi: 10.1111/j.1365-2672.2012.05344.x). In a more recent study, 135 human-derived Lactobacillus and Bifidobacterium strains have been screened for their ability to produce GABA from its precursor monosodium glutamate. Fifty eight strains were found able to produce GABA. The most efficient GABA-producers were Bifidobacterium strains together with some species of Lactobacilli (L. plantarum, L. brevis). About the issue raised by the Reviewer that the compared strains in the study by Barrett et al. are too limited to state that L. Brevis and B. Dentium are the most effective producers of GABA, we agree with the Reviewer and therefore, we have changed the statement accordingly.
- L124: How were the experimental groups assigned?
The study was designed as a double-blind, sham-controlled, randomized clinical trial. A detailed explanation about the randomization, masking and blinding procedures has been added in the Materials and Methods section (4.2 Study Design).
- L536: just to be clear, was alcohol intake allowed? Since nicotine was allowed, were all forms (smoked, skin absorption…) accepted?
As we affirmed in the Methods section, during the entire study, all participants were prescribed a hypocaloric diet, to standardize the food kinds in the 3 treatment groups. The daily dietary intake included approximately 45% to 50% calorie intake from carbohydrate, up to 30% of calories from fat, and 20% to 25% of calories from protein. Obviously, alcohol consumption was not expected in the diet.
Nicotine consumption was not considered an exclusion criteria, however we preferred not to discontinue the nicotine during the study, in order to avoid neurotransmitter imbalances (e.g. serotonin, glutamate, endogenous opiates) in smoker patients, due to nicotine withdrawal
- L542: how did the authors control the use of prebiotics naturally occurring on the diet (eg. Granola, oat, other fibres…)?
As stated in the previous point, all participants in the 3 treatment groups were prescribed a hypocaloric Mediterranean diet. The total carbohydrate amount included about 20-25 gr/day of fibres. According to the Italian Society of Human Nutrition (SINU) guidelines, the fibres amount has been computed as 16 gr/1000 Kcal/day.
The use of probiotics and yogurt was not allowed during the study period.
- L618: how the fecal samples were collected and how long it took to be frozen?
Fecal samples were collected in sterile containers before and after 15 dTMS sessions and were stored at -20°C until analysis. Fecal samples were frozen within 3 hours from the collection, and if this was not possible, they were temporarily stored at 2-8 °C until delivery.
- Table 1: Leptin and Epinephrine have a higher SEM compared to the other two groups, and specially the leptin Sham group has a mean that is almost the double that HF. Could you explain why of this variability?
Leptin is an adipose-derived peptide hormone in direct proportion to amount of body fat. However, leptin levels could be affected by other conditions. For example, higher leptin levels have been found to be associated with increased substances craving, insulin resistance and systemic inflammation (TNF-α, IL6), with important physiological implications. Likewise, catecholamine levels have been shown to be potentially affected by multiple factors such as age, sex, body mass index (BMI), hypertensive disease, sodium intake, diet. Higher SEMs for these biomarkers reflect their considerable variability. However, comparisons among groups obtained with ANOVA, followed by Bonferroni post-hoc test for normally distributed values, and with Kruskal-Wallis for non-normally distributed values, did not find significant differences both in the leptin and epinephrine levels among the 3 groups at baseline.
- Are these and other data normal? Was performed a power analysis to identify the correct number of subjects per group? Why HF group has 9 subjects while LF and Sham have 6 and 7, respectively?
This aspect has been clarified both in Materials and Methods and Discussion sections.
- L146-152: Where are these data in the manuscript? Were remaining phyla not statistically significant or not assessed?
The data concerning the phyla, genera and all the variables analyzed in the study were reported in a Table in the Supplementary Material (Table S1). The remaining phyla were not analyzed, as their abundance was negligible compared to that of the two major phyla: Bacteroidetes and Firmicutes.
- Figure 2 legend and description: I found a bit difficult to follow the statistical analysis on figure 2. It is not clear when the data refers to comparison within treatment or before-after treatment.
The figure legend has been modified according to the Reviewer’s suggestions.
- Figure 3 needs to be redesigned.
As suggested by the Reviewer, Figure 3 has been redesigned.
Round 2
Reviewer 1 Report
Thank you for the opportunity to review the modified version. Again, this manuscript is well written and interesting. Although I am still concerned about the statistical power, I recommend it to be published.
Author Response
We really appreciated the Reviewer's interest toward our paper and his/her constructive comments aimed at improving the quality of our paper. We are aware that a limit of the study is the small sample size, however we added a power calculation in the statistical method section considering that the aim of the study was to evaluate the effects of a 5-week treatment on gut microbiota composition looking at the correlations between microbiota variations and metabolic, neuro-hormonal changes.
The main endpoint was considered the correlation between the norepinephrine change and Bacteroides genus variation, and a sample size of 23 achieves 80% power to detect a difference of 0.45 between the null hypothesis correlation of 0.25 and the alternative hypothesis correlation of 0.70 using a two-sided hypothesis test with a significance level of 0.05.
We also included a comment in the discussion section, stating that this is an exploratory study and no formal statistical hypotheses on the correlation between gut microbiome composition changes and other biomarker variations were pre-specified to determine the sample size.
In fact, this is the first study in which the composition of the gut microbiota was studied before and after a treatment with deep Transcranial Magnetic Stimulation (dTMS), not only in individuals with obesity, but also in the whole TMS-treated population. Therefore, for the calculation of the sample size we had no previous data to refer to, then we relied on a robust hypothesis.
Our hypothesis was based on the ability of the dTMS to modulate both the Central Nervous System (CNS) and the Autonomic Nervous System (ANS), through the Central Autonomic Network (CAN), and to affect the brain-gut communication pathways. For the post hoc power statistical calculation, we decided to correlate the main neurotransmitter used by the sympathetic nervous system and involved in the CAN, the norepinephrine, with the Bacteroides genus, which represents the most substantial portion of the mammalian gastrointestinal microbiota, where it plays a fundamental role in processing of complex molecules to simpler ones in the host intestine. Furthermore, Bacteroides genus belonging to Bacteroidetes phylum was found to be significantly associated with body weight changes, supporting previous evidence of a positive correlation with weight loss. Therefore, we hypothesized that it was the bacterial genus that was most likely affected by dTMS-treatment and most suggestive of an improvement in the gut microbiota composition.
A comment on the analysis regarding the power of the study has been added in the Discussion section (page 14, lines 518-527). Revisions in the manuscript have been highlighted, using the "Track Changes" function in Microsoft Word.